# Properties of CuFeS_2_/TiO_2_ Nanocomposite Prepared by Mechanochemical Synthesis

**DOI:** 10.3390/ma15196913

**Published:** 2022-10-05

**Authors:** Erika Dutkova, Matej Baláž, Nina Daneu, Batukhan Tatykayev, Yordanka Karakirova, Nikolay Velinov, Nina Kostova, Jaroslav Briančin, Peter Baláž

**Affiliations:** 1Department of Mechanochemistry, Institute of Geotechnics, Slovak Academy of Sciences, Watsonova 45, 04001 Košice, Slovakia; 2Advanced Materials Department, Jožef Stefan Institute, 1000 Ljubljana, Slovenia; 3Department of General and Inorganic Chemistry, Al-Farabi Kazakh National University, Almaty 050040, Kazakhstan; 4Institute of Catalysis, Bulgarian Academy of Sciences, 1113 Sofia, Bulgaria

**Keywords:** CuFeS_2_, TiO_2_, mechanochemical synthesis, CuFeS_2_/TiO_2_ nanocomposite, photocatalytic activity

## Abstract

CuFeS_2_/TiO_2_ nanocomposite has been prepared by a simple, low-cost mechanochemical route to assess its visible-light-driven photocatalytic efficiency in Methyl Orange azo dye decolorization. The structural and microstructural characterization was studied using X-ray diffraction and high-resolution transmission electron microscopy. The presence of both components in the composite and a partial anatase-to-rutile phase transformation was proven by X-ray diffraction. Both components exhibit crystallite size below 10 nm. The crystallite size of both phases in the range of 10–20 nm was found and confirmed by TEM. Surface and morphological properties were characterized by scanning electron microscopy and nitrogen adsorption measurement. Scanning electron microscopy has shown that the nanoparticles are agglomerated into larger grains. The specific surface area of the nanocomposite was determined to be 21.2 m^2·^g^−1^. Optical properties using UV-Vis and photoluminescence spectroscopy were also investigated. CuFeS_2_/TiO_2_ nanocomposite exhibits strong absorption with the determined optical band gap 2.75 eV. Electron paramagnetic resonance analysis has found two types of paramagnetic ions in the nanocomposite. Mössbauer spectra showed the existence of antiferromagnetic and paramagnetic spin structure in the nanocomposite. The CuFeS_2_/TiO_2_ nanocomposite showed the highest discoloration activity with a MO conversion of ~ 74% after 120 min irradiation. This study has shown the possibility to prepare nanocomposite material with enhanced photocatalytic activity of decoloration of MO in the visible range by an environmentally friendly manner

## 1. Introduction

In recent years, water pollution caused by dyes has received widespread attention. Photocatalysis is a promising method for dyes degradation as an alternative to conventional ones. In the photocatalysis, a variety of semiconductors such as TiO_2_, ZnO, WO_3_, etc., have been used, and the photocatalytic activity of selected metal oxide photocatalysts was recently reported in a review paper [1]. Nanocrystalline TiO_2_ produced through microwave irradiation as the most promising photocatalyst was investigated by Nunes et al. [2]. However, it can only absorb photons in the UV region, which comprises 5% of the solar spectrum. One way to improve the photocatalytic activity of TiO_2_ is coupling with other semiconductor, e.g., CdS, ZnS, CuInS_2_, CuInSe_2_, CuFeS_2_, etc. [3,4,5,6]. Hybrid nano-structured systems as visible-light photocatalysts may be developed due to the attractive TiO_2_ with its superlative photo-activity, low cost and toxicity, and convenient band gap energy (3–4 eV) [7].

CuFeS_2_ is a ternary I-III-VI semiconductor with narrow direct band gap for solar light harvesting and possesses chemical stability and less toxicity, and has garnered recent attraction due to its high photo-stability and high photo-conversion efficiency [8]. Coupling of CuFeS_2_ semiconductor (small bulk band gap is 0.6 eV) with TiO_2_ (higher bulk band gap is 3.2 eV) can improve the photocatalytic activity due to a change in the rate of carriers recombination from the so-produced hybrid systems. CuFeS_2_/TiO_2_ hybrid material could be an extraordinary alternative for dye degradation in near-visible or visible-light ranges of spectrum.

Many synthetic approaches have been reported for the preparation of TiO_2_-coated ternary sulphide semiconductors (CuInS_2_, CuInSe_2_, etc.). The preparation of magnetic TiO_2_-coated CuFeS_2_ nanomaterial by a combination of solvothermal and wet-impregnation methods for photocatalysis applications was investigated in [9]. Mechanochemical synthesis is a powerful method for synthesis of a wide range of materials applying the high-energy milling to impact and speed up chemical reactions [10,11]. This route is simple, solvent-free, and reproducible, and also the mechanochemical treatment could be readily scaled up [12,13]. We have already applied the mechanochemical synthesis for the preparation of other metal sulphide-TiO_2_ nanocomposites with good photocatalytic properties [14,15].

The CuFeS_2_/TiO_2_ nanocomposite synthesized by mechanochemical synthesis was not investigated until now. The innovation of this study is the simple non-conventional mechanochemical method of CuFeS_2_/TiO_2_ nanocomposite synthesis for a very short time, at ambient pressure and temperature. Moreover, higher photocatalytic activity of the mechanochemically synthesized CuFeS_2_/TiO_2_ nanocomposite in degradation of methyl orange azo dye was discovered.

## 2. Materials and Methods

### 2.1. Materials

For the mechanochemical synthesis of the CuFeS_2_/TiO_2_ nanocomposite, we used mechanochemically prepared CuFeS_2_, which was synthesized from elemental copper (99.7%, Aldrich, Germany), iron (extra pure, Merck, Germany), sulphur (99%, Ites, Slovakia) and commercially available TiO_2_ P25 powder (Degussa, Netherland) (75% anatase and 25% rutile).

### 2.2. Mechanochemical Synthesis of CuFeS_2_/TiO_2_ Nanocomposite

CuFeS_2_ was previously synthesized in a planetary mill Pulverisette 6 (Fritsch, Idar-Oberstein, Germany) by milling of 1.73 g of elemental copper, 1.52 g iron and 1.75 g of sulphur. The milling was carried out at 550 rpm using a tungsten carbide milling chamber (250 mL in volume) and 50 balls (10 mm in diameter), composed of the same material, during 60 min in an argon atmosphere according to Equation (1) and the procedure described in [16].

CuFeS_2_/TiO_2_ nanocomposite in the molar ratio 1:4 (chosen based on the previous results [14,15]) was prepared by the co-milling of 1.824 g of previously synthesized CuFeS_2_ and 3.176 g of commercial TiO_2_. The co-milling of CuFeS_2_ and TiO_2_ was carried out in a planetary mill Pulverisette 6 (Fritsch, Idar-Oberstein, Germany) according to Equation (2) in an argon atmosphere for only 30 min. The 250 mL tungsten carbide milling chamber with 50 tungsten carbide balls (360 g), 10 mm in diameter, was used. The rotational speed of the planet carrier was 500 rpm. The ball-to-powder ratio was 72:1.

Preparation of the CuFeS_2_/TiO_2_ nanocomposite can be described by the subsequent reactions and is displayed in Figure 1:Cu + Fe + 2S → CuFeS_2_(1)
CuFeS_2_ + TiO_2_ → CuFeS_2_/TiO_2_(2)

### 2.3. Characterization Methods

X-ray diffraction (XRD) measurements were performed using a D8 Advance diffractometer (Bruker, Germany) equipped with a two-circle goniometer, CuK radiation (40 kV, 40 mA), a secondary graphite monochromator, and a scintillation detector. All samples were scanned from 20° to 80° with steps 0.03° and 12 s counting time. Diffracplus Eva software was used for phase analysis according to the ICDD-PDF2 database. The Rietveld refinement was performed using a TOPAS Academic software [17,18].

Microstructural characteristics of the samples were further investigated by transmission electron microscopy (TEM). A small amount of sample was dispersed in pure ethanol and ultrasonicated for 5 min. Then, a droplet of the suspension was settled into a lacey carbon-coated nickel grid and dried. The sample was carbon-coated to prevent charging under the electron beam. TEM analyses were performed on a 200 kV microscope JEM 2100 (JEOL, Japan) with LaB_6_ electron source and equipped with energy-dispersive X-ray spectrometer (EDXS) for chemical analysis.

The values of the specific surface area were acquired by using a NOVA 1200e Surface Area & Pore Size Analyzer (Quantachrome Instruments, Boynton Beach, FL, USA).

Morphology was studied using a field emission–scanning electron microscope FE–SEM, Mira 3 (Tescan, Brno, Czech Republic) coupled with an EDS analyzer (Oxford Instruments, Oxford, UK).

Absorption spectra were measured using a UV-Vis spectrophotometer Helios Gamma (Thermo Electron Corporation, Warwickshire, UK) by dispersing the sample in absolute ethanol by ultrasonic stirring.

Photoluminescence (PL) spectra were recorded on a photon-counting spectrofluorometer PC1 (ISS, San Antonio, TX, USA) with a photoexcitation wavelength of 325 nm. A 300-W xenon lamp was used as the excitation source. The photoexcitation and photoemission slit widths were 1 and 2 mm, respectively. The powder samples were suspended in absolute ethanol.

The EPR spectra were recorded using a JEOL JES-FA 100 EPR spectrometer (JEOL, Japan). The EPR spectrometer operated in X-band, and it is furnished with a standard TE_011_ cylindrical resonator. The samples were settled in special quartz tubes and settled in the cavity centre. The EPR measurements were performed at room temperature at the following instrumental settings: modulation frequency of 100 kHz, microwave power 1 mW, and amplitude of the magnetic field modulation of 0.2 mT.

The room temperature Mössbauer spectra were measured with a Wissel (Wissen-schaftliche Elektronik GmbH, Starnberg, Germany) electro-mechanical spectrometer working in a constant acceleration mode. A ^57^Co/Rh (activity = 50 mCi) source and α-Fe standard were applied. The isomer shift (δ), quadrupole splitting (Δ), quadrupole shift (2ε), magnetic hyperfine field (Bhf), line widths (Γexp), and relative weight (G) of partial components in the spectra as parameters of hyperfine interaction were found using CONFIT2000 software [19].

The photocatalytic activity in the reaction of Methyl Orange (MO) dye decoloration was measured under visible-light illumination (λ ≥ 420 nm). The photocatalytic experiments were performed using a semi-batch photocatalytic reactor equipped with a magnetic stirrer, similarly to our previous work [14]. MO dye was used as a model pollutant with the concentration of 10 mg/L. The suspension was prepared by adding the investigated sample (100 mg) to 100 mL of MO solution. By means of optical absorption spectroscopy (M501 Single Beam scanning UV/Visible Spectrophotometer, CamSpec Cambridge Spectrophotometers Co., UK), the MO degree of discoloration was monitored. The suspended sample was magnetically mixed in the dark for 30 min to ensure an adsorption–desorption equilibrium. Then, the suspension was illuminated by visible light. All experiments were carried out at a constant mixing rate of 400 rpm at room temperature. By observation of the changes of the main absorbance peak at λ = 463 nm, the concentration of MO during the photocatalytic reaction was identified.

## 3. Results and Discussion

### 3.1. Structural and Microstructural Characterization

The XRD measurements were performed to study the phase composition of the prepared sample. The XRD pattern of CuFeS_2_/TiO_2_ nanocomposite synthesized by mechanochemical route is given in Figure 2.

The X-ray diffraction pattern is dominated by signals corresponding to TiO_2_ and CuFeS_2_, as expected. During the mechanochemical treatment, TiO_2_ anatase modification has been almost completely transformed to rutile, as the main peak belonging to anatase (located at 25.4°) is barely visible. In addition, the Rietveld refinement has estimated its content to be only around 1%. The anatase-to-rutile transformation upon milling is in accordance with the previously reported results [14]. The estimated content of the rutile phase is around 70%, while that of chalcopyrite CuFeS_2_ is between 15% and 20%. It was found that, in addition to chalcopyrite, pyrite (FeS_2_) is also present in the reaction mixture (its most intense diffraction peak is located at 56.0°, and its content is between 10 and 15%). This phase is most probably formed as a side-product during the chalcopyrite synthesis from elemental precursors and has also been reported earlier [20,21,22]. In addition, a small amount of bornite (Cu_5_FeS_4_) is also probably present.

The estimated crystallite sizes of both main phases are similar, namely, that of rutile TiO_2_ is 9.3 ± 0.6 nm and that of chalcopyrite is 9.3 ± 0.8 nm. The obtained crystallite size of chalcopyrite is much smaller than the 38 nm reported in [16]; however, in this case, it might decrease during the second stage (co-milling with TiO_2_) due to the formation of nanocomposite structure. In the only report on the CuFeS_2_/TiO_2_ nanocomposite [9], the crystallite size of chalcopyrite is reported to be 38.3 nm and that of TiO_2_ 7.3 nm, the latter value being in accordance with our observation.

TEM was further used to study the microstructural characteristics of CuFeS_2_/TiO_2_ nanocomposite. The results are shown in Figure 3.

Low-magnification TEM image of the CuFeS_2_/TiO_2_ nanocomposite (Figure 3a) illustrates that the sample is composed of up to micron large agglomerates, which is common in mechanochemical synthesis. Selected area electron diffraction (SAED) patterns (Figure 3b) produce diffraction rings with interplanar distances belonging to chalcopyrite (Ccp, red arcs) and rutile (Rt, white arcs), thus confirming that the agglomerates are composed of randomly oriented nanocrystallites of predominantly these two phases. EDXS analysis (Figure 3c) confirmed that Cu, Fe and S from chalcopyrite and Ti and O form rutile are present in the sample, as expected. High-resolution image (Figure 3d) shows that the sample is composed of fine crystallites with sizes ranging from 10 to 20 nm; some smaller and few larger crystallites were also observed in this and other parts of the sample. Crystallites of both phases, rutile (Rt) and chalcopyrite (Ccp) were identified in the HRTEM image based on the difference in the d-values of the most intense (characteristic) lattice planes (110)Rt and (112)Ccp, which measure 3.4824 Å and 3.0385 Å, respectively. The difference was confirmed by FFT pattern (Figure 3e), calculated from the squared area indicated in Figure 3b. In this area, two crystallites with visible lattice spacings are observed. Diffraction spots from both grains yield spots with different d-values, i.e., different diameters of the rings centered around the central spot, as indicated on the FFT pattern (red for Ccp and white for Rt). In addition, the lattice spacings were measured directly from HRTEM images by measuring 10 lattice spacings (to reduce the error) in each grain, as seen in the intensity profiles shown in Figure 3f. It can be observed that 10 lattice spacings in both grains yield different distances, 3.24 and 3.04, which can be attributed to rutile and chalcopyrite, respectively.

### 3.2. Surface and Morphological Characterization

The specific surface area (S_A_) values belong to some of the most notable characteristics of the samples prepared by milling. The S_A_ of pure CuFeS_2_, from which studied nanocomposite was synthesized, is 2.7 m^2·^g^−1^, as was published in a paper by Dutkova et al. [16]. In the present case, the significant amount of TiO_2_ results in much higher specific surface area value, namely, 21.2 m^2·^g^−1^. The S_A_ of pure TiO_2_ P25 is 28.7 m^2·^g^−1^. The specific surface area of CuFeS_2_:TiO_2_ 1:1 composite prepared by solvothermal/impregnation approach reported in [9] was 73.7 m^2·^g^−1^. A significantly higher specific surface area in the mentioned study is most probably a result of the used experimental conditions, under which the porous structure can be formed. On the contrary, during mechanochemical synthesis, the powders are all the time being bombarded with balls, so there is no space for the well-ordered porous structure to form.

The morphology of the prepared nanocomposite was obtained by means of scanning electron microscopy (SEM). The SEM micrograph of the synthesized CuFeS_2_/TiO_2_ nanocomposite is displayed in Figure 4. The SEM image manifests that the sample is composed of fine nanoparticles forming closely compacted irregular agglomerates with irregular size and shape. The agglomerates exhibit the size in micrometers; however, smaller units with the sizes in the nanometer range (50–250 nm in sizes) can be clearly distinguished. The smaller nanoparticles may be traced on the surface of the larger ones as well as they are distributed between them.

The EDS mapping was done in order to investigate the distribution of individual elements, and the results are included in Figure 5. The EDS layered image comprehensive of all the elements of CuFeS_2_/TiO_2_ nanocomposite is illustrated in Figure 5A. The individual energy-dispersive X-ray spectroscopy (EDS) mappings for Cu, Fe, S, Ti and O distribution are shown in Figure 5B–F. From the results, it can be found that in CuFeS_2_/TiO_2_ nanocomposite the observed elements are dispersed equally.

### 3.3. Optical Properties

The optical absorption properties of TiO_2_ and the mechanochemically synthesized of CuFeS_2_/TiO_2_ nanocomposite (Figure 6a) were investigated by UV–Vis spectroscopy. By displaying (αhν)^2^ against (hν) and extrapolating the slope in the band edge section to zero, the optical band gaps were found as visualized in Figure 6b. The optical band gap of TiO_2_ was determined as 3.34 eV, which is blue-shifted in comparison with its bulk analogue (3.2 eV). Mechanochemically synthesized CuFeS_2_/TiO_2_ nanocomposite exhibits the narrow band gap (2.75 eV). The optical band of previously synthesized CuFeS_2_ nanoparticles was calculated as 1.05 eV [16], which is larger than the band gap of the bulk CuFeS_2_ (0.6 eV) resulted from the nano-size effect as a consequence of milling. The observed band gap value of the nanocomposite is in the space of range for pure CuFeS_2_ and TiO_2_, as was expected due to mixing of both semiconducting components; consequently, a slight red shift may occur for TiO_2_ and a slight blue shift may occur for CuFeS_2_. The coupling of CuFeS_2_ with the TiO_2_ will result in a red shift of absorption value due to formation of new energy state levels, as was reported in the case of other semiconductors coupled with TiO_2_ [23]. The decrease in band gap value is beneficial for the photocatalytic application of TiO_2_ in the visible region.

TiO_2_ and CuFeS_2_/TiO_2_ nanocomposite samples were photo-excited by irradiation with light at a wavelength of 325 nm. It matches to the photon energy of 3.81 eV, which is absorbed by exciting transition of valence band electrons to the conduction band. The PL spectra are represented in Figure 7. The PL spectrum of TiO_2_ shows three maxima, two of them in the violet region at 398 nm (3.1 eV), 425 (2.9 eV) and one in the blue region at 467 nm (2.6 eV). The emission peak at about 398 nm is associated to the emission of band gap transition [24] and it originates from the radiative recombination of excitons [25]. The emission peak at 467 nm is attributed to the defects present in the sample, especially the oxygen-ion vacancies, which ensure acceptor levels near the conduction-band edge [26]. In the case of CuFeS_2_/TiO_2_ nanocomposite, the broad emission bands distributed in the wavelength 350–500 nm are composed of two very weak sub-bands centered at 375 nm and 467 nm, but the shape of the emission curve becomes irregular with Gaussian distribution in contrast to TiO_2_. PL intensity of TiO_2_ after coupling with CuFeS_2_ is quenched. The copper in CuFeS_2_ probably captures electrons from TiO_2_ and reduces the recombination rate. It is in agreement with the results noted in the paper by Kang et al. [9]. The higher number of defects in the sample are the consequence of mechanical energy input during the solid-state reaction.

An EPR study was performed in order to characterize the oxidation state of copper and iron in CuFeS_2_ and CuFeS_2_/TiO_2_ nanocomposite. The EPR spectra are displayed in Figure 8. In the spectrum of the CuFeS_2_ sample, a broad signal with g factor 2.1922 can be interpreted as a superposition from two signals; the first one resulting from distorted octahedrally coordinated Cu^2+^ ions and the second one due to Fe^3+^ ions. The copper species could be dimeric or monomeric Cu, and the Fe ions have octahedrical coordination. However, because of overlapping of the signals, it is not possible to exactly identify the species.

The EPR spectrum of mechanochemically synthesized CuFeS_2_/TiO_2_ nanocomposite consists of superposition of two overlapping EPR lines with g≈2.2 and g = 2.1636. These lines are better resolved than at pure CuFeS_2_ and most probably are related with two different paramagnetic species. The first one is due to paramagnetic Cu^2+^ ions. According to literature data, the second line is related to the presence of Fe^3+^ ions in Fe_2_O_3_-type clusters. In addition, at low magnetic field, a line at g value 4.34 due to isolated rhombic Fe^3+^ in anatase is recorded [27,28]. Therefore, the EPR analysis has found two types of paramagnetic ions in the nanocomposites.

Mössbauer spectra of CuFeS_2_ and CuFeS_2_/TiO_2_ nanocomposite are described in Figure 9 and the calculated hyperfine values are summarized in Table 1. The experimental Mössbauer spectrum of the CuFeS_2_ sample is a superposition of sextet and doublet components. A model containing two sextets and one doublet was applied to achieve the best fitting results. The parameters of sextet Sx1 (Table 1) correspond well to the parameters of tetrahedrally coordinated Fe^3+^ ions in antiferromagnetic chalcopyrite [29,30]. The differences in the parameters of Sx2 such as the lower value of the effective magnetic field, as well as line broadening (*Γ_exp_* = 0.82 mm/s), compared to the parameters of Sx1, could be explained by the presence of chalcopyrite crystallites with smaller size.

The doublet component has parameters characteristic of Fe^3+^ ions and may be due to chalcopyrite particles with superparamagnetic behavior. It should be noted that a doublet with similar parameters could be also assigned to octahedrally coordinated Fe^2+^ in pyrite (FeS_2_) structure [31]. The presence of pyrite in CuFeS_2_ was confirmed in our previous papers [21,22]. The experimental Mössbauer spectrum of CuFeS_2_/TiO_2_ sample is also a combination of sextet and doublet components. The parameters of the sextet in CuFeS_2_/TiO_2_ spectrum are close to the parameters of Sx2 in CuFeS_2_ spectrum, which was assigned to smaller-size chalcopyrite crystallites. The doublet component has parameters close to those of the doublet in CuFeS_2_, but its relative weight is significantly higher. Therefore, it can be assumed from Mössbauer analysis that, as a result of mechanochemical synthesis of CuFeS_2_/TiO_2_ composite, the chalcopyrite crystallites with large size characterized by Sx1 in the CuFeS_2_ sample are absent in CuFeS_2_/TiO_2_ sample, and the amount of superparamagnetic chalcopyrite particles has been significantly increased in CuFeS_2_/TiO_2_. Moreover, an additional phase of pyrite has been formed, as confirmed by XRD.

### 3.4. Photocatalytic Properties

The photodegradation effect of Methyl Orange (MO) on TiO_2_, CuFeS_2_ and CuFeS_2_/TiO_2_ samples can be pursued based on its concentration decrease with the increasing time under visible-light illumination. Figure 10 shows the photocatalytic degradation efficiency C/C_0_ of the dye as a function of time. All experiments were performed under natural pH and at room temperature.

Photocatalytic activity of mechanochemically synthesized CuFeS_2_/TiO_2_ nanocomposite shows an enriched performance, in comparison with the pure commercial TiO_2_ P25 Degussa and the pure mechanochemically prepared CuFeS_2_. The pure TiO_2_ P25 Degussa shows only ~ 4% of discoloration of Methyl Orange under visible-light irradiation, while the mechanochemically synthesized CuFeS_2_ reveals ~ 52% of discoloration under similar degradation conditions (Figure 10a). The mechanochemically synthesized CuFeS_2_/TiO_2_ nanocomposite showed the highest discoloration activity with a MO conversion of ~ 74% after 120 min irradiation (Figure 10a). The concentration of Methyl Orange without catalyst was stable under visible-light irradiation (Figure 10a). The pseudo-first-order kinetics of Methyl Orange degradation over various photocatalysts are depicted in Figure 10b. The apparent pseudo-first-order rate constants *k* are 0.0005, 0.0073, and 0.0120 min^−1^ for the TiO_2_, CuFeS_2_, and CuFeS_2_/TiO_2_ catalysts, respectively (Figure 10b), which once confirms that the mechanochemically prepared CuFeS_2_/TiO_2_ composite exhibits the highest photocatalytic activity. The mechanochemically synthesized CuFeS_2_/TiO_2_ nanocomposite exhibited high photocatalytic activity for Methyl Orange and stable performance without obvious decrease after three runs (Figure 10c), showing a promising application for the photocatalytic decolorization of dyes. A slight decrease in the photocatalytic activity during four cycles may be due to the surface neutralization of the catalyst by Methyl Orange photodegradation products. As the photocatalytic removal of various impurities (most often dyes) from water with the use of various nanocomposites is currently one of the top research topics, on the basis of the achieved results the application of synthesized nanocomposite for removal of others dyes from water may be further studied in the future.

The superior photocatalytic activity owing to the separation efficiency of photo-induced electron-hole pairs resulted from the heterojunctions between the interface of TiO_2_ and CuFeS_2_. The following empirical Equation (3) was used to calculate the position of the edge of the TiO_2_ valence band at the point of zero charge [32]:E_CB_ = *χ* − E^e^ − 0.5 Eg (3)
where E_CB_ is the CB potential, χ is the absolute theoretical electronegativity for the TiO_2_ semiconductor 5.90 eV [33,34], E^e^ is the free electron energy of TiO_2_ about 4.5 eV on the hydrogen scale, Eg is the band gap, for titanium oxide 3.34 eV, which was determined experimentally by UV-visible spectroscopy. The positions of the edges of the VB and CB bands of TiO_2_ nanoparticles were determined as 2.91 and –0.44 eV, respectively, and the positions of CuFeS_2_ were +1.45 and +0,4 eV, respectively [9,35]. The presented diagram in Figure 11, describing the band structure of the CuFeS_2_/TiO_2_ nanocomposite, shows the redox potentials of exemplary reactions that occur during photocatalysis in an aqueous medium, where the presence of dissolved oxygen molecules is also taken into account. On Figure 11 is a proposed diagram of the stages of photogenerated electron/hole transfer under visible-light irradiation at the CuFeS_2_/TiO_2_ catalyst interface. When illumination is done with visible light with a wavelength of more than 420 nm, only CuFeS_2_ is able to absorb photons, which leads to the transfer of electrons from the VB to the TiO_2_ CB. Owing to the electrostatic field at the junction, photoexcited electrons can easily pass into CB, while holes remain in CuFeS_2_, thereby increasing the lifetime of photogenerated electrons and holes. As a result, separated electrons and holes can initiate reduction and oxidation reactions with O_2_ and H_2_O molecules adsorbed on the catalyst surface. The potential of VB CuFeS_2_ (+1.45 eV) is higher than that of the •OH/H_2_O pair (+2.32 eV), and holes in VB CuFeS_2_ cannot react with H_2_O to form •OH radicals. However, these holes can oxidize Methyl Orange to Methyl Orange+. The accumulated electrons on CB CuFeS_2_ can easily react with O_2_ and release •O_2_ radicals into aqueous solution. These radicals can directly oxidize the dye. Nevertheless, the electron holes formed on the CuFeS_2_ surface play the main role in the mechanism of Methyl Orange photodegradation.

## 4. Conclusions

In this work, a CuFeS_2_/TiO_2_ nanocomposite has been prepared by a simple, low-cost mechanochemical route to assess its visible-light driven photocatalytic efficiency in azo dye decolorization. The XRD analysis of the CuFeS_2_/TiO_2_ nanocomposite, prepared by dry milling in a planetary ball mill, showed phase transition of anatase into rutile during milling. Both components exhibit crystallite size below 10 nm. Mechanochemically synthesized CuFeS_2_/TiO_2_ nanocomposite exhibits the narrow band gap (2.75 eV) and stronger absorption band from the UV to Vis region. The results showed the existence of antiferromagnetic and paramagnetic spin structure in nanocomposite. The enhanced photocatalytic activity of decoloration of MO in aqueous solutions of the synthesized CuFeS_2_/TiO_2_ nanocomposite in the visible region of the spectrum is because of the forceable separation of photo-excited electrons and holes—the charge carriers between the CuFeS_2_ and the TiO_2_ attached phases. The advantage of this study is a sustainable production of photocatalytically active composite material. Challenges for the future may be to properly investigate the ratio between the two phases CuFeS_2_ and TiO_2_ to achieve the best results.

## Figures and Tables

**Figure 1 materials-15-06913-f001:**
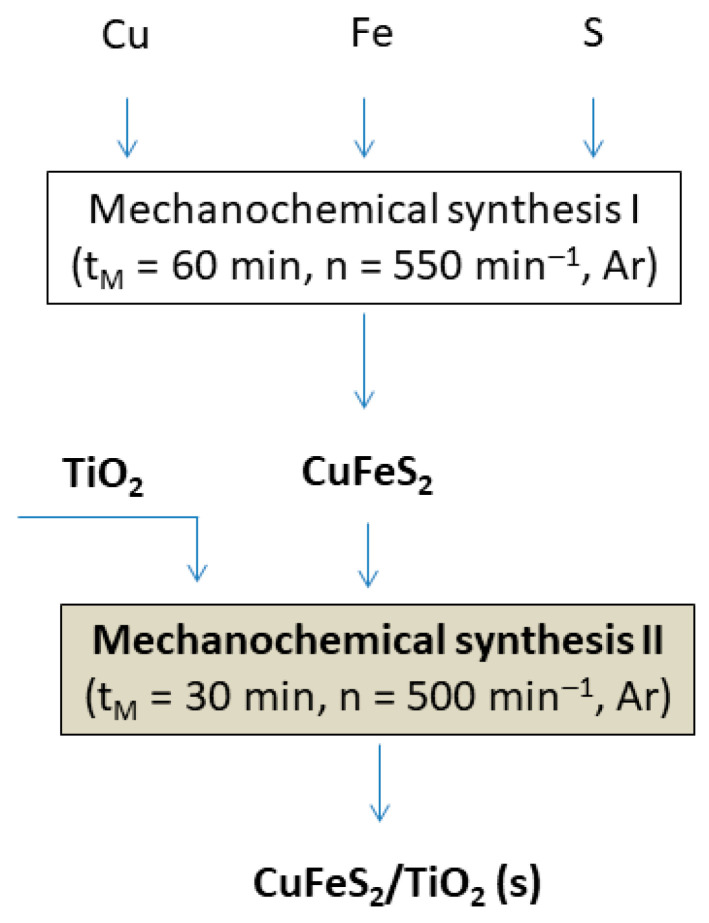
Flowsheet of the CuFeS_2_/TiO_2_ nanocomposite preparation.

**Figure 2 materials-15-06913-f002:**
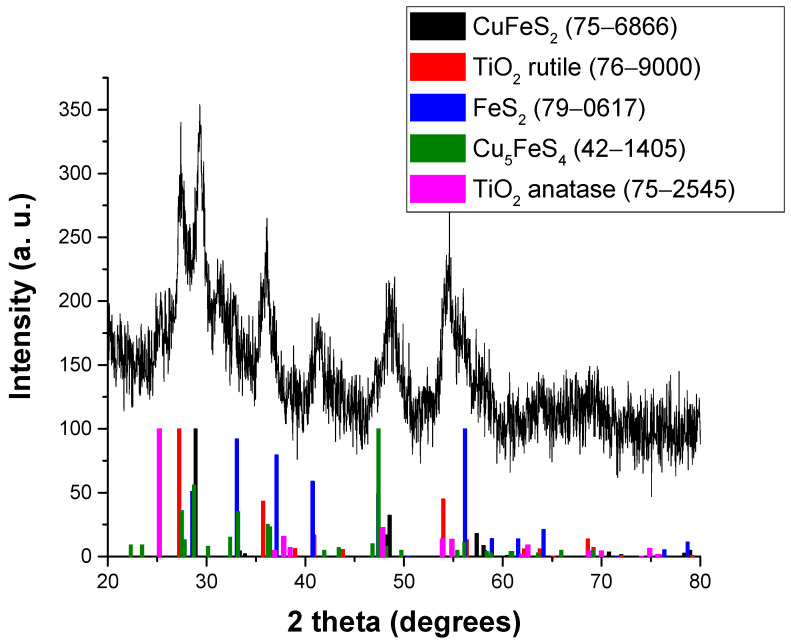
XRD pattern of CuFeS_2_/TiO_2_ nanocomposite.

**Figure 3 materials-15-06913-f003:**
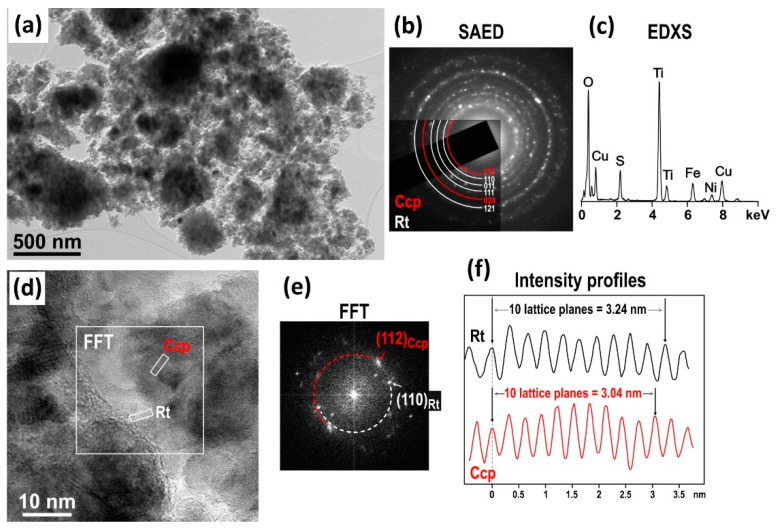
TEM analysis of CuFeS_2_/TiO_2_ nanocomposite: (**a**) TEM image at low magnification with indexed SAED pattern that contains diffraction rings of rutile (Rt) and chalcopyrite (Ccp); and EDXS spectrum shows the presence of Cu, Fe, S, Ti and O in the sample. The SAED pattern (**b**) and EDXS analyses (**c**) included circular areas with approximately 500 nm diameter. (**d**) High-resolution TEM image of the sample. The presence of Rt and Ccp grains was confirmed by FFT analysis (**e**) and from intensity profiles (**f**) taken across the crystallites with visible lattice spacings. See text for more detailed description.

**Figure 4 materials-15-06913-f004:**
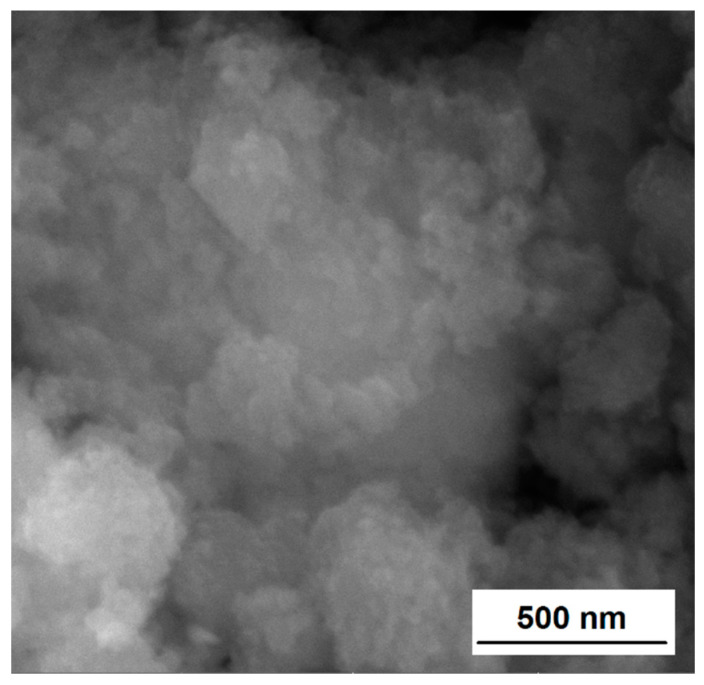
SEM of CuFeS_2_/TiO_2_ nanocomposite.

**Figure 5 materials-15-06913-f005:**
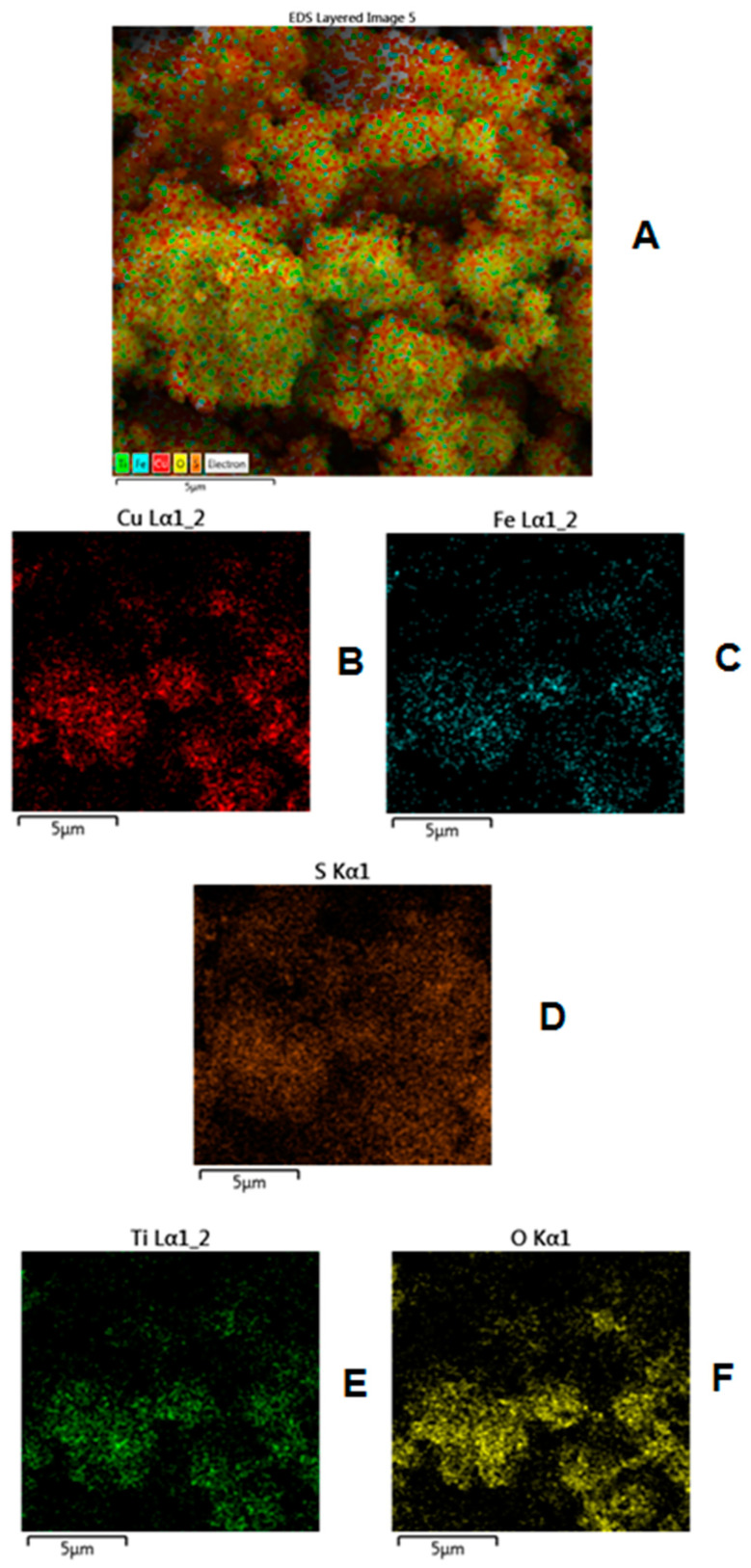
EDS mapping of CuFeS_2_/TiO_2_ nanocomposite. (**A**) EDS layered image; (**B**) elemental mapping for Cu distribution; (**C**) elemental mapping for Fe distribution; (**D**) elemental mapping for S distribution; (**E**) elemental mapping for Ti distribution; (**F**) elemental mapping for O distribution.

**Figure 6 materials-15-06913-f006:**
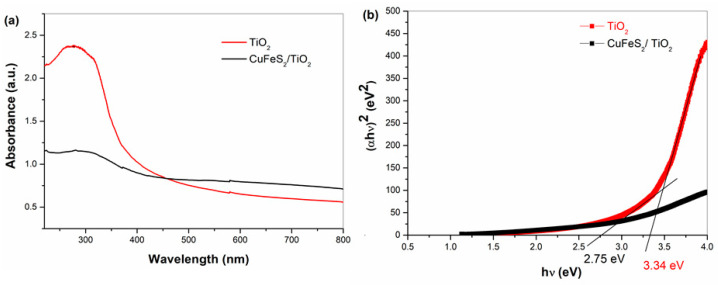
(**a**) UV–Vis spectra and (**b**) Tauc plots for TiO_2_ and CuFeS_2_/TiO_2_ nanocomposite.

**Figure 7 materials-15-06913-f007:**
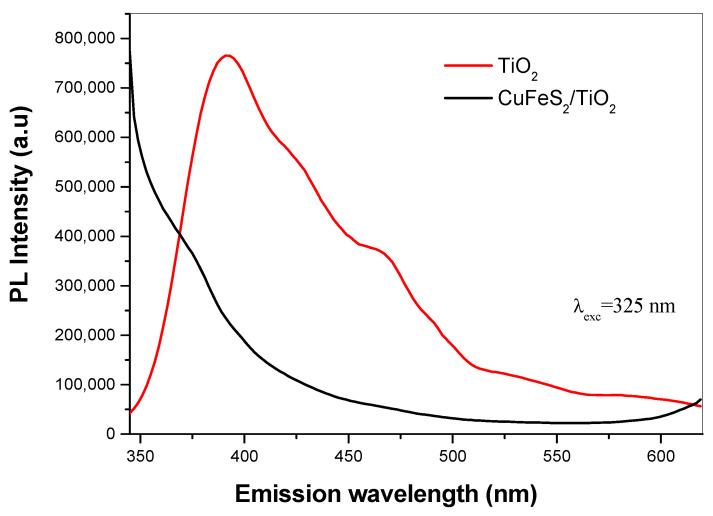
PL spectra of TiO_2_ and CuFeS_2_/TiO_2_ nanocomposite.

**Figure 8 materials-15-06913-f008:**
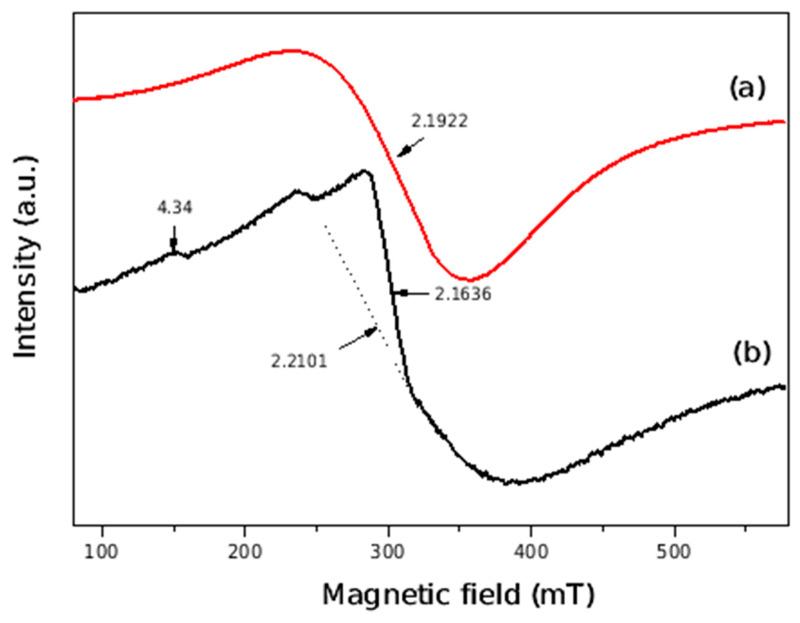
EPR spectra of the mechanochemically synthesized (**a**) CuFeS_2_ and (**b**) CuFeS_2_/TiO_2_ nanocomposite.

**Figure 9 materials-15-06913-f009:**
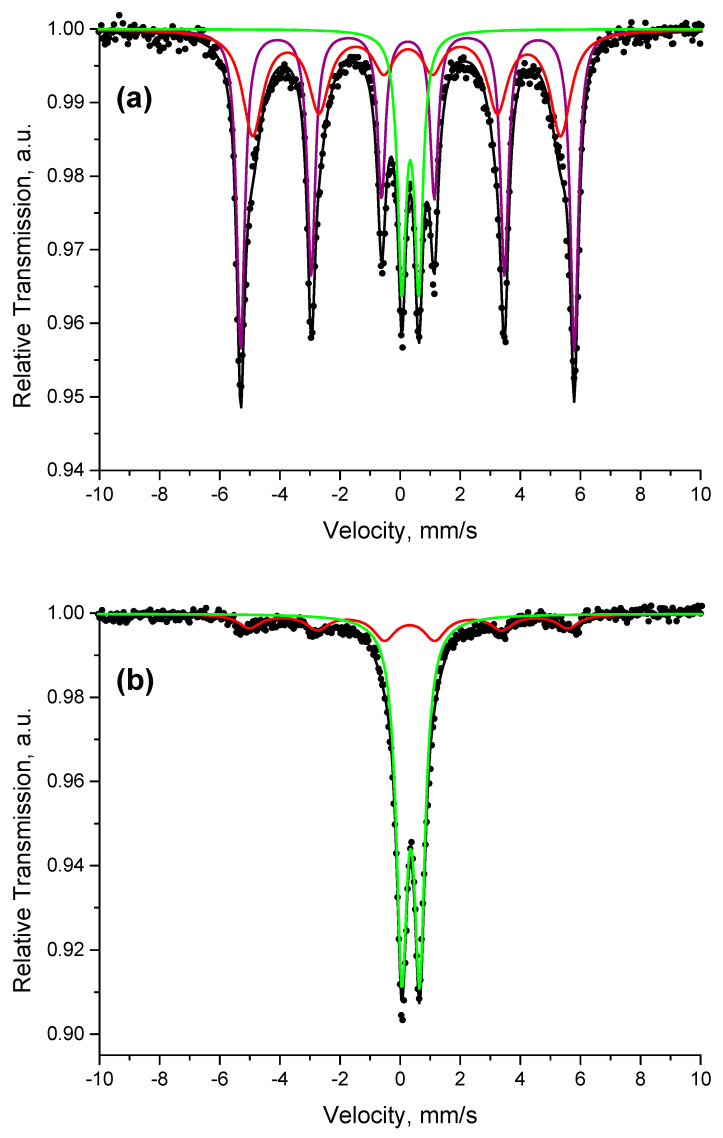
Mössbauer spectra of (**a**) CuFeS_2_ and (**b**) CuFeS_2_/TiO_2_ nanocomposite.

**Figure 10 materials-15-06913-f010:**
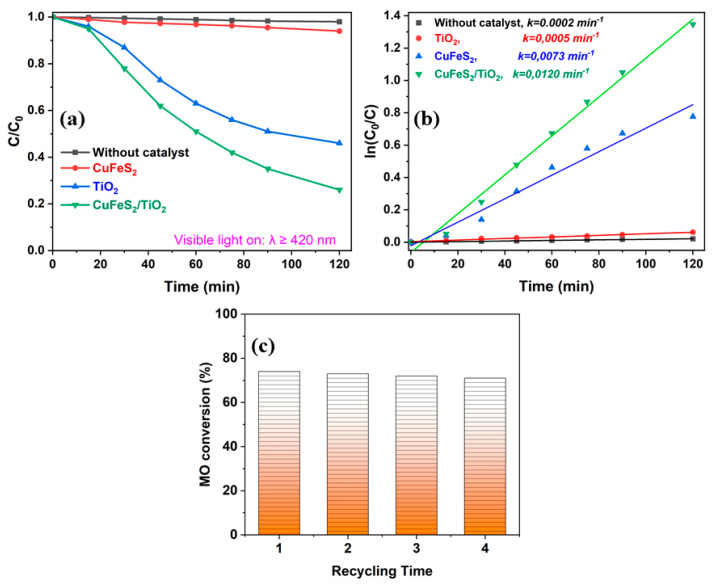
Curves of photocatalytic decomposition (**a**) of MO on commercial TiO_2_ P25 Degussa, mechanochemically synthesized CuFeS_2_, mechanochemically synthesized CuFeS_2_/TiO_2_ and without a catalyst under visible-light illumination, determination of the rate constants of photocatalytic decomposition (**b**) and photocatalytic stability of CuFeS_2_/TiO_2_ for 4 cycles (**c**).

**Figure 11 materials-15-06913-f011:**
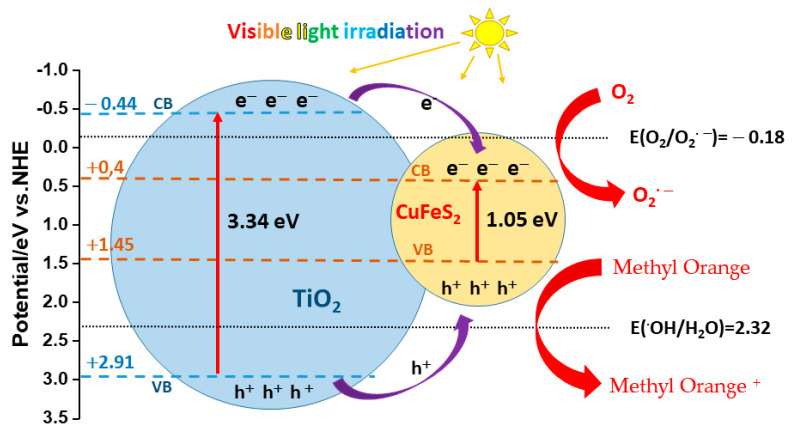
Schematic illustration of delocalization of charge carriers and the formation of reactive regions on the catalyst surface under visible light illumination.

**Table 1 materials-15-06913-t001:** Mössbauer values of CuFeS_2_ and CuFeS_2_/TiO_2_ nanocomposite.

Sample	Components	*δ*, mm/s	Δ *(2ε)*, mm/s	*B_hf_*, T	*Γ_exp_*, mm/s	G, %
CuFeS_2_	Sx1 Sx2 Db	0.25 0.25 0.34	−0.01 −0.05 0.56	35.5 31.8 -	0.32 0.82 0.34	47 36 17
CuFeS_2_/TiO_2_	Sx Db	0.30 0.35	−0.03 0.60	32.9 -	0.88 0.44	24 76

## Data Availability

The data presented in this study are available on request from the corresponding author.

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
