# Peer review of "Properties of CuFeS2/TiO2 Nanocomposite Prepared by Mechanochemical Synthesis"

_materials, 2022, doi:10.3390/ma15196913_

Round 1

Reviewer 1 Report

The work reports the production of CuFeS2/TiO2 nanocomposite by a mechanochemical route to assess its visible-light driven photocatalytic efficiency in Methyl Orange azo dye decolorization.

In the introduction, please rephrase the sentence:

There is only one report for the preparation of TiO2-coated CuFeS2 nanomaterials [7]. 50 In this paper, CuFeS2@TiO2 magnetic nanoparticles prepared by a combination of sol- 51 vothermal and wet-impregnation methods for photocatalysis applications were studied. This is not a scientific way to compare your study. You must include some important references, also published on MPDI:

Metal oxide-based photocatalytic paper: A green alternative for environmental remediation, Catalysts 202111(4), 504; https://doi.org/10.3390/catal11040504

Photocatalytic TiO2 Nanorod Spheres and Arrays Compatible with Flexible Applications, Catalysts 20177(2), 60; https://doi.org/10.3390/catal7020060

In the introduction, it is not explained the photocatalytic process.

What is the state-of-the-art regarding the mechanochemical process for producing TiO2? It is not presented, and must be included.

In materials section, it is not clear how and the order that the powder is introduced. Can you please be more specific and explain the complete process? It is hard to follow the preparation of the material, if it is not explained.

In the results, how can you prove that all the pristine precursors were converted to the desired phase?

It is not clear by XRD that it has been converted.

The estimated crystallite size of both main phases are similar, namely that of rutile 161 TiO2 is 9.3 ± 0.6 nm and that of chalcopyrite is 9.3 ±0.8 nm. Please explain how it has been calculated and add references.

On figure 3, please add the FFT images and index the patterns. It is imperative to prove the lattice spacing. It is wrong to present like that, since the lattice spacing is really close and hard to identify it (d(112)cp = 0.3038 nm) and rutile ((110)rt = 0.3248 nm).

I tis important to measure particle size on TEM images not on SEM. And consider at least 30 particles to properly measure it.

Regarding the photocatalysis, can you please indicate the lost of photocatalytic activity with cyclings? Indicate on the image.

Can you please discuss why it gets worst with the cycling tests?

Can you compare your photocatalytic activity with other materials?

And justify the photocatalytic behaviour observed.

The english must be improved and the problems must be addressed.

Author Response

We have carefully re-read the manuscript and tried to response to the reviewer comments.

Reviewer 2 Report

This manuscript reported a simple method for preparing of CuFeS2/TiO2 nanocomposite with enhanced photocatalytic activity. The most advantage of this work is the potential of large-scale preparation of photocatalysis. The following issues should be solved before it could be considered for publication.

1). XRD, TEM and SEM of CuFeS2 and TiO2 should be given and discussed;

2). UV-Vis spectra, Tauc plots and PL of CuFeS2 should be shown;

3). EPR and Mössbauer spectrum of TiO2 should be shown;

4). Blank tests of the Methyl Orange without photocatalyst should be shown.

Author Response

We have carefully re-read the manuscript and tried to correct all the mistakes and to response to reviewer comments.

Reviewer 3 Report

In the present contribution Dutkova et al. described a preparation of novel nanocomposite containing copper, sulfur, iron and titanium and its efficient use in photocatalytic removal of methyl orange azo dye from water.

As the photocatalytic removal of various impurities (most often dyes) from water with the use of various nanocomposites is currently one of the top research topics, I consider the research undertaken to be absolutely justified. The results are presented clearly, the work does not have any substantive shortcomings.

Taking into account the above considerations, I recommend publishing this paper in Materials.

Author Response

Many thanks for the comments. We have re-read our article once again and tried to correct all the mistakes found. 

Reviewer 4 Report

1. Grammatical and typo errors must be rectified to improve the quality of the manuscript.

2. At 500 rpm of mill speed, there will be a chance of contamination of tungsten in to CuFeS2 nanocomposite, because a big challenge in ball milling is the contamination and oxidation. There will be traces of oxidation even after using argon as milling atmosphere. How did authors managed to get rid of oxidation and contamination.

3. What was the ball to powder ratio used for mechanochemical synthesis of the nanocomposites?

4. Some of the peaks in XRD are shifted compared to the indexing, what is the reason for that?

5. As per the figure 1, authors milled total of 90 minutes (60 min for preparing CuFeS2 and 30 min to prepare CuFeS2/TiO2 composite), but in the line 81, author mentioned only 30 minutes, it is very confusing for the readers.

6. It is recommended to provide a mechanism diagram for better understanding of photodegradation process.

7. Similarity index is more than 35%, this is a very serious problem. Revise it accordingly

Author Response

We have re-read our article once again and tried to correct all the mistakes found.

Round 2

Reviewer 1 Report

The manuscript has been improved, however in my opinion it makes no sense to make a photocatalyst that could overcome TiO2 limitation of it's absorption under UV light and present results of photodegradation under UV. Takes credit from the work.

Author Response

We agree with the Reviewer. In the experimental part, it is written that photocatalysis was carried out under UV and visible irradiation. But the presented results are only under irradiation with visible light. Therefore, we corrected the experimental part and left there that the experiment was carried out using only visible irradiation. We would like to apologize for this mistake.

Reviewer 2 Report

The authors have solved all the comments and suggestions from reviewer, it could be accepted as it is.

Author Response

Many thanks for your acceptation.

Reviewer 4 Report

The authors revised the manuscript as per my suggestion, so I am recommending the acceptance of the manuscript in its current form.

Author Response

Many thanks to the reviewer for the acceptance.

Reviewer 5 Report

OK, I think the paper could be accepted.

Author Response

(The authors gave the same response as above.)
